# Autophagic cell death is dependent on lysosomal membrane permeability through Bax and Bak

Jason Karch[1,2], Tobias G Schips[1], Bryan D Maliken[1], Matthew J Brody[1], Michelle A Sargent[1], Onur Kanisicak[1], Jeffery D Molkentin[1,2]*

[1]Department of Pediatrics, Cincinnati Children's Hospital Medical Center, University of Cincinnati, Ohio, United States; [2]Howard Hughes Medical Institute, Cincinnati, United States

**Abstract** Cells deficient in the pro-death Bcl-2 family members Bax and Bak are known to be resistant to apoptotic cell death, and previous we have shown that these two effectors are also needed for mitochondrial-dependent cellular necrosis (Karch et al., 2013). Here we show that mouse embryonic fibroblasts deficient in *Bax/Bak1* are resistant to the third major form of cell death associated with autophagy through a mechanism involving lysosome permeability. Indeed, specifically targeting Bax only to the lysosome restores autophagic cell death in *Bax/Bak1* null cells. Moreover, a monomeric-only mutant form of Bax is sufficient to increase lysosomal membrane permeability and restore autophagic cell death in *Bax/Bak1* double-deleted mouse embryonic fibroblasts. Finally, increasing lysosomal permeability through a lysomotropic detergent in cells devoid of *Bax/Bak1* restores autophagic cell death, collectively indicting that Bax/Bak integrate all major forms of cell death through direct effects on membrane permeability of multiple intracellular organelles.

DOI: https://doi.org/10.7554/eLife.30543.001

*For correspondence:
jeff.Molkentin@cchmc.org

Competing interests: The authors declare that no competing interests exist.

## Introduction

Programmed cell death type II, also known as autophagic cell death, is defined as cell death in the presence of lysosomes (*Schweichel and Merker, 1973*). Morphologically, autophagic cell death is characterized by vacuolization of the cytoplasm but with a lack of chromatin condensation as seen in apoptosis, or with a lack of organelle swelling as observed in necrosis (*Kroemer et al., 2009*). The existence of an autophagic cell death pathway as a bona fide active mechanism for cellular killing is somewhat controversial (*Kroemer and Levine, 2008*). Fundamentally, the autophagic process is meant for cellular survival in times of nutrient deprivation or during other stresses that lead to cellular wasting (*Gozuacik and Kimchi, 2007*). Induction of autophagy results in the selective breakdown of cellular organelles and other proteins to redistribute energy to essential cellular processes as a last line of defense to maintain cellular viability (*Gozuacik and Kimchi, 2007*). However, if the nutrient deprivation persists this adaptive process transitions to a more deadly phase in which there is irreversible degradative consequences and progression to a form of cell death that is morphologically distinct from apoptosis or necrosis.

A distinct mechanistic feature that separates autophagic cell death from the other forms of cell death is the change in activity and accumulation of lysosomes (*Kroemer et al., 2009*). Lysosomal-dependent cell death was first proposed in 1970's based on visual rupture of this organelle (*Aits and Jäättelä, 2013*). Lysosomal membrane permeabilization (LMP) and rupture has also been implicated as a concurrent process during apoptosis (*Johansson et al., 2010*). During LMP cathepsins are released into the cytosol of the cell where they induce caspase-dependent apoptosis or

directly lead to necrosis (*Boya and Kroemer, 2008*). Depending on the stress and cell type many molecules have been proposed to influence LMP but the exact mechanism is unknown (*Boya and Kroemer, 2008*). One proposed mechanism is through the pro-apoptotic Bcl-2 family member Bax (*Kågedal et al., 2005*; *Feldstein et al., 2006*; *Oberle et al., 2010*). For example, *Bax* and *Bak1* double-knockout (DKO) mouse embryonic fibroblasts (MEFs) persist in serum-free and nutrient poor media while wild-type (WT) MEFs die (*Wei et al., 2001*). However, *Bax/Bak1* DKO MEFs are also resistant to both apoptotic and regulated necrotic cell death so the mechanistic basis for how Bax/ Bak might normally permit a regulated form of autophagic cell death is unclear (*Karch and Molkentin, 2015*). Here, we determined that autophagic cell death is dependent on Bax or Bak localized to the lysosome in their monomeric states where they permit LMP. In the absence of Bax/Bak, lysosomes are impermeable and have enhanced acidity and fail to rupture during a macro-autophagic event, while restoring LMP in *Bax/Bak1* DKO MEFs rescues serum starvation-induced autophagic cell death.

## Results

### Bax and Bak are required for autophagic cell death

*Bax/Bak1* DKO MEFs are highly resistant to serum starvation-induced cell death (*Wei et al., 2001*). We confirmed this previous result and also examined whether reconstitution with either Bax or Bak was sufficient to restore this type of cell death in DKO MEFs (*Figure 1A*). We also observed a similar level of protection in DKO MEFs subjected to amino acid-free media when compared to WT MEFs (*Figure 1—figure supplement 1A*). Indeed, expressing either Bax or Bak1 in the DKO background was sufficient to restore serum starvation-induced cell death, suggesting a role in autophagic killing (*Figure 1A*). We confirmed that serum starvation elicited an autophagic response with an LC3-GFP marker expressed by adenoviral-based transduction into WT and DKO MEFs (*Figure 1B*). LC3-GFP showed a punctate expression pattern after 24 hr serum starvation in both WT and DKO cell lines (*Figure 1B* ). Western blot analysis of autophagy related proteins also showed induction over 4 days of serum starvation, such as a shift from LC3-I to LC3-II, greater Lamp1 levels and cleavage of cathepsin D into its active form (*Figure 1C*). Transmission electron microscopy (EM) showed an accumulation of autophagy vacuoles and expanded lysosomes after two days of starvation in WT and DKO MEFs. However vacuoles in the DKO MEFs were smaller and less frequent compared with WT MEFs, andDKO MEFs with starvation uniquely showed dark punctate bodies in the cytoplasm (*Figure 1D*). Following four days of starvation WT MEFs broadly displayed loss of plasma membrane integrity whereas the plasma membrane of the DKO MEFs remained intact (*Figure 1D*).

### Autophagic cell death is a unique form of death

To determine if autophagic cell death is dependent upon the apoptotic process or mitochondrial-dependent necrotic cell death we examined how serum starvation-induced cell death is affected by caspase inhibitors or by desensitization of the mitochondrial permeability transition pore (MPTP), respectively. During apoptosis Bax/Bak oligomerize to form large pores in the outer mitochondrial membrane to enable cytochrome-*c* release and formation of the apoptosome, which mediates caspase activation and apoptotic cell killing (*Korsmeyer et al., 2000*). Here, WT MEFs treated with a pan-caspase inhibitor (Z-VAD-FMK) showed no resistance to serum starvation-induced cell death, suggesting that this form of autophagic-regulated killing was caspase independent (*Figure 2A*). Bax/Bak are also critical regulators of MPTP-dependent necrosis (*Karch et al., 2013*). To determine if serum starvation-induced cell death also requires a regulated necrotic process through the MPTP we subjected *Ppif* null MEFs (*Ppif* encodes the gene for cyclophilin D) to serum starvation, although no change in levels of cell death were observed suggesting that autophagic cell death proceeds in a manner independent of the MPTP (*Figure 2B*).

Bax/Bak oligomerization is indispensable for apoptotic and necroptotic forms of regulated cell death (*Antonsson et al., 2000*; *Karch et al., 2015*). However, we previously showed that the monomeric states of Bax or Bak are sufficient to mediate increased permeability of the outer mitochondrial membrane to permit MPTP-dependent, regulated necrotic killing of cells (*Karch et al., 2013*).

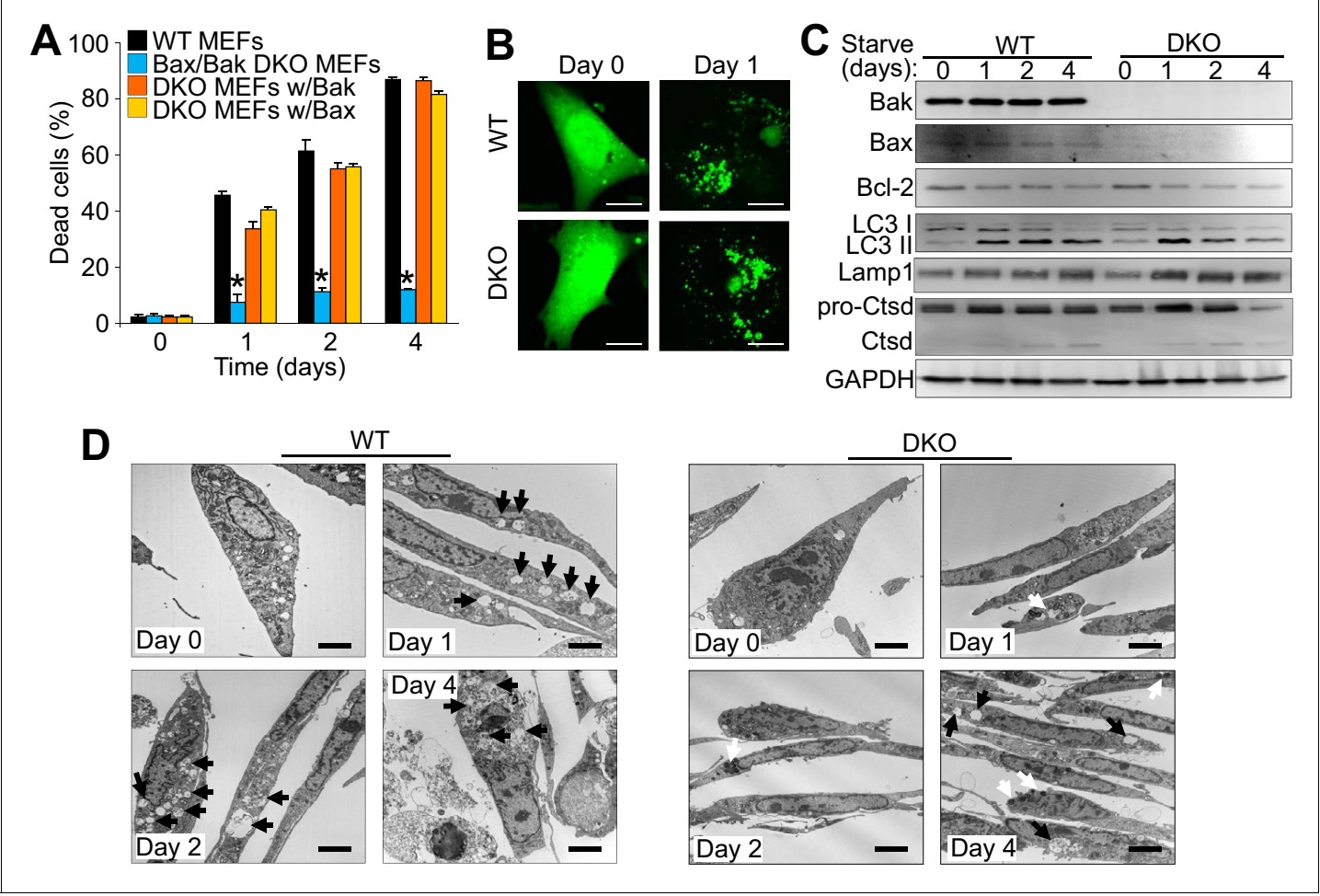

**Figure 1.** The expression of Bax or Bak are required for autophagic cell death. (**A**) Wild type (WT) mouse embryonic fibroblasts (MEFs), *Bax/Bak1* double knockout (DKO) MEFs, and DKO MEFs stably expressing Bax or Bak1 were subjected to serum starvation for the indicated time points and cell death was measured by loss of membrane integrity. (**B**) Confocal images of WT and DKO MEFs that were infected with an LC3-GFP (green fluorescence) adenovirus and were untreated (0) or subjected to serum starvation for 1 day. The figure shows punctate LC3 at 1 day in both types of MEFs. Scale bars = 5 µm. (**C**) Western blot analysis of Bax, Bak, Bcl-2, LC3, Lamp1, Cathepsin D (Ctsd), and GAPDH (control) in WT and DKO MEFs during serum starvation for the indicated time points in days. (**D**) Transmission electron microscopic (EM) images (10,000x) of WT and DKO MEFs at baseline or serum starved up to 4 days. Black arrows highlight autophagic-like vacuoles and lysosomes. White Arrows show punctate bodies that are specific to the DKO cells with starvation. Scale bars = 2.5 µm. All assays are representative of three independent experiments. Averages are shown and error bars represent the standard error of the mean. *p<0.05 vs 0 time point. Statistical significance was determined by students t-test.
DOI: https://doi.org/10.7554/eLife.30543.002

The following source data and figure supplement are available for figure 1:

**Source data 1.** Raw western gel images for *Figure 1C*.
DOI: https://doi.org/10.7554/eLife.30543.004

**Figure supplement 1.** The Expression of Bax or Bak are required for autophagic cell death induced by amino acid deprivation.
DOI: https://doi.org/10.7554/eLife.30543.003

Here we used two different oligomerization-deficient mutant versions of Bax reconstituted at a similar level to the WT control line within the background of DKO MEFs (*Figure 2—figure supplement 1A*) (*Hoppins et al., 2011*). Remarkably, while DKO MEFs were resistant to serum starvation-induced cell death, reconstitution with Bax 63-65A or Bax 92-94A restored autophagic cell death to levels observed with WT Bax reconstitution (*Figure 2C*). However, using an active-Bax specific antibody showed that this factor was not directly activated in serum-depleted WT MEFs, and Bax translocation was also undetectable in serum-starved DKO MEFs expressing a Bax-GFP fusion protein, although as a control apoptosis induction with staurosporine caused abundant translocation and Bax

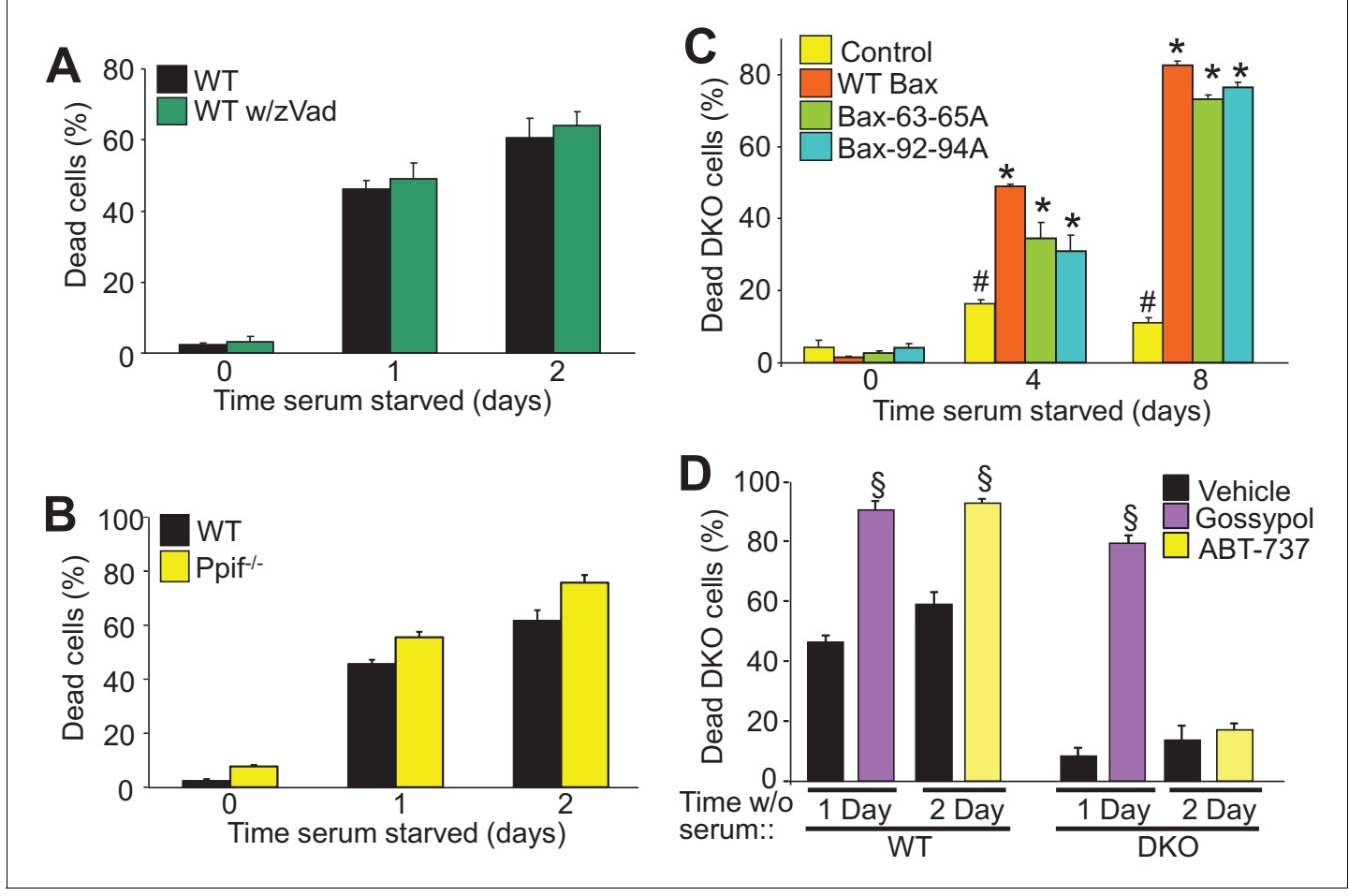

**Figure 2.** Autophagy can culminate in a unique form of cell death. (**A**) WT MEFs treated with or without a pan caspase inhibitor (zVad) and subjected to serum starvation for 1 or 2 days. Cell death was measured by plasma membrane rupture. (**B**) WT and *Ppif* null MEFs were subjected to serum starvation and cell death was assessed as in 'A'. (**C**) WT MEFs and DKO MEFs stably expressing oligomeric dead mutants of Bax (Bax63-65A or Bax 92-94A) or WT Bax were subjected to serum starvation and cell death was measured by loss of plasma membrane integrity for the days shown. (**D**) WT and DKO MEFs were treated with or without BH3-mimtics (ABT-737 or Gossypol) and were subjected to serum starvation. Cell death was assessed as in 'A'. All assays are an average three independent experiments. The error bars represent the standard error of the mean. #$p<0.05$ vs WT Bax. *$p<0.05$ vs 0 time point. §$p<0.05$ vs vehicle. Statistical significance was determined by students t-test.

DOI: https://doi.org/10.7554/eLife.30543.005

The following source data and figure supplements are available for figure 2:

**Figure supplement 1.** Bax/Bak activation, oligomerization, and translocation does not occur during autophagic cell death.

DOI: https://doi.org/10.7554/eLife.30543.006

**Figure supplement 1—source data 1.** Raw western gel images for *Figure 2—figure supplement 1A and C*.

DOI: https://doi.org/10.7554/eLife.30543.007

activation (*Figure 2—figure supplement 1B,D*). Finally, Bak oligomerization was not detected in serum starved WT MEFs but it was with staurosporine (*Figure 2—figure supplement 1C*). These results indicate that Bax/Bak activation and oligomerization are not required for serum starvation-induced cell death.

WT and DKO MEFs were also treated with the BH3-memtic compounds gossypol and ABT-737 to further explore the mechanism of Bax/Bak-dependent cell death with serum starvation. Both of these compounds antagonize and inhibit the anti-apoptotic Bcl-2 family members (such as Bcl-2 and Bcl-xl). However, gossypol is unique in that it elicits a conformational change in Bcl-2 that makes it pro-apoptotic like Bax/Bak, while ABT-737 simply blocks the activity of the anti-apoptotic Bcl-2 family members (*van Delft et al., 2006*; *Lei et al., 2006*). Here, treatment with either of these agents

enhanced the kinetics of cell death in WT MEFs, but only gossypol was able to restore autophagic killing in DKO MEFs (*Figure 2D*). The observed effect on WT MEFs simply indicates that augmenting the activity of Bax/Bak, or artificially creating more 'Bax/Bak-like' activity can augment autophagic programmed killing. However, because gossypol but not ABT-737 restored autophagic killing in DKO MEFs, it further indicates that this form of cell death requires the expression of Bax/Bak.

## DKO MEFs have increased lysosomal acidity and decreased permeability

Closer inspection of serum-starved WT and DKO MEFs by transmission EM showed greater intracellular vesicular fragmentation in WTs with 1 and 2 days of starvation along with the appearance of electron dense vesicles in the DKOs (*Figure 3A*). To assess the identity of these vesicles live cell imaging was performed with LysoTracker red (lysosomes) and rhodamine-123 to assess mitochondrial membrane potential. We observed little to no difference in the mitochondria membrane potential between starved WT and DKO MEFs (green stain) but there was a dramatic difference in the fluorescence intensity of LysoTracker red (*Figure 3B*). LysoTracker red partitions to acidic vacuoles and its fluorescent intensity reflects accumulation in such structures, which are typically lysosomes or autolysosomes (*Wubbolts et al., 1996*). After 2 days of serum starvation DKO MEFs showed 3-fold greater LysoTracker red intensity than comparably starved WT MEFs (*Figure 3C*), which was also observed in DKO MEFs subjected to amino acid free media (*Figure 1—figure supplement 1B*). To measure LMP directly we treated starved and LysoTracker red loaded WT and DKO MEFs with bafilomycin A1 to inhibit the vacuolar-type $H^+$-ATPase, which is responsible for the acidification of the lysosome and autolysosome (*Figure 3D,E*). If there is ongoing LMP, bafilomycin A1 treatment will lead to loss of accumulated LysoTracker red signal, but if LMP is very low, fluorescence will remain high. Indeed, bafilomyocin A1 treatment showed loss of red fluorescence in WT MEFs while DKO MEFs retained fluorescence indicating that loss of Bax/Bak no longer permits LMP (*Figure 3D,E*). Together these data suggest that Bax/Bak are required for LMP during autophagic stress.

To further investigate the mechanism of autophagic cell death due to Bax and Bak, cytosolic pH and total cathepsin B activity were measured during serum starvation. Twenty-four hours of serum starvation in WT MEFs significantly lowered cytosolic pH compared with unstarved, suggesting that LMP and lysosomal rupture was occurring during autophagic conditions (*Figure 3F*). However, serum starvation in DKO MEFs showed no reduction in cytosolic pH suggesting inhibition of LMP and lysosomal rupture in the absence of Bax and Bak (*Figure 3F*). Associated with reduced cytosolic pH is an aggregation of activated cathepsin B, as seen in serum starved WT MEFs over 3 days but not in DKO MEFs (*Figure 3G*). These results suggest that in the absence of Bax and Bak the cytosol remains neutral pH, likely because lysosomes and autolysosomes remained intact.

## Lysosomal-only Bax is sufficient to restore autophagic cell death in DKO MEFs

Bax and Bak regulate cell death, in part, by localization to the mitochondria where they generate either apoptotic pores or increase the general permeability characteristics of the outer membrane for MPTP-dependent necrotic cell death (*Wei et al., 2001*; *Karch et al., 2013*). However, Bax and Bak can also localize to the lysosome where they can further enhance apoptotic cell death through an unknown mechanism (*Kågedal et al., 2005*; *Feldstein et al., 2006*; *Oberle et al., 2010*). Indeed, sub-cellular fractionation of WT MEFs both in the fed and starved state, showed increased Bax and Bak within the lysosome compartment that was coincident with lysosomal-associated membrane protein 1 (Lamp1) and devoid of mitochondrial proteins for oxidative phosphorylation (*Figure 4A*). To better visualize lysosomes at the ultrastructural level we generated a fusion construct between Bax and mini-singlet oxygen generator (miniSOG). MiniSOG-EM provides visualization of a protein by transmission EM without the loss of visualization of some membrane-based structures that can plague Immuno-gold antibody based imaging (*Shu et al., 2011a*). For this experiment DKO MEFs were infected with a recombinant adenovirus expressing a miniSOG-Bax fusion protein and 24 hr later imaged for SOG activity to show localization of Bax. As expected, miniSOG-Bax protein was found in the cytoplasm and was also associated with the mitochondrial membrane (yellow arrows), as well as within lysosomal/autophagosomal membranes (*Figure 4B*, pink arrowheads). These results

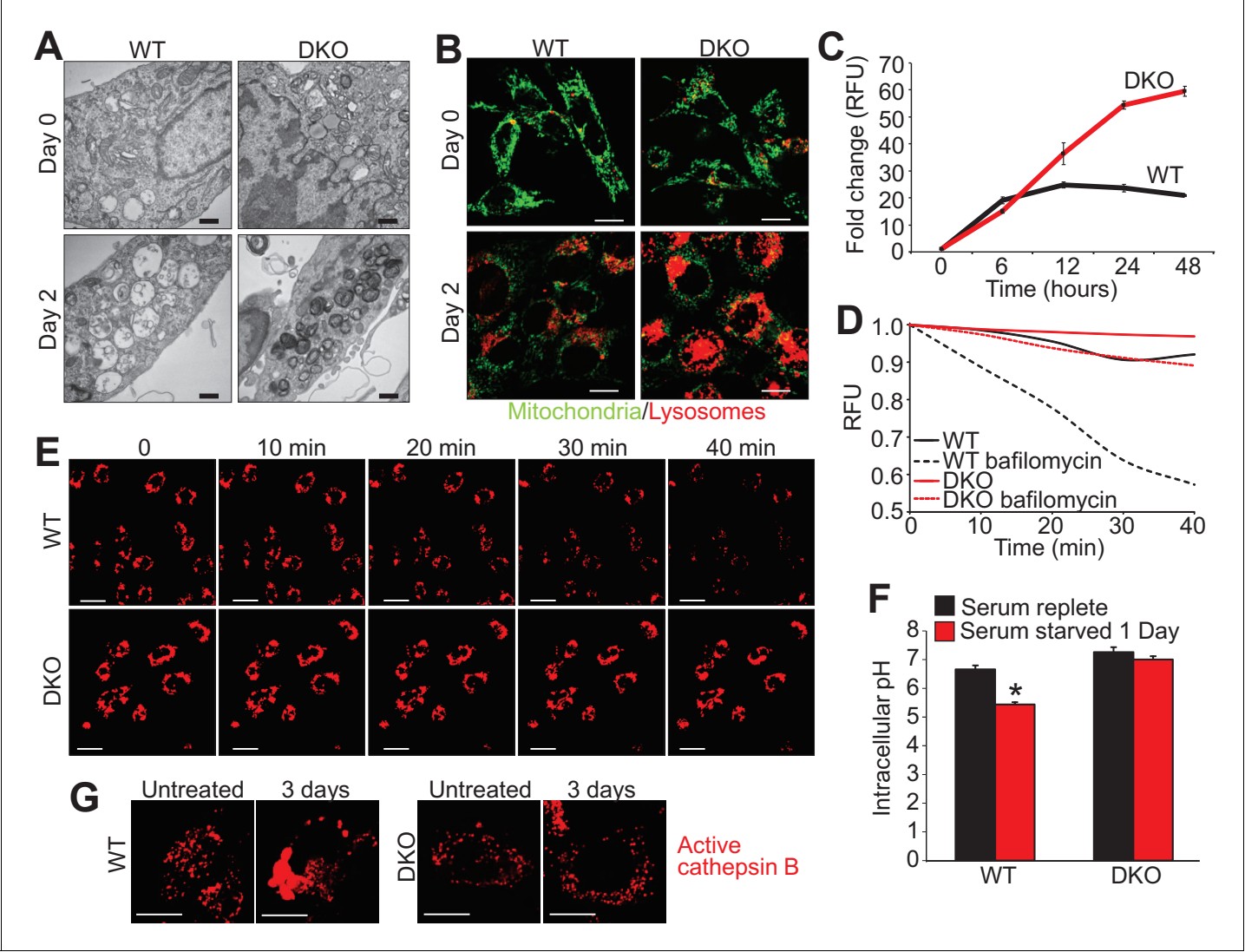

**Figure 3.** Loss of *Bax* and *Bak1* maintains lysosomal integrity. (**A**) Transmission EM images (30,000x) of WT and DKO MEFs subjected to serum starvation for the specified time points. Autolysosomes and translucent vesicles are noticeable in the serum-starved WT MEFs but not in the DKO MEFs. Scale bars = 1 μm. (**B**) Confocal images of WT and DKO MEFs subjected to serum starvation for the indicated times and labeled with Rhodamine 123 (green) to measure mitochondrial membrane potential and LysoTracker red (red) to label lysosomes. Scale bars = 5 μm. (**C**) Time course of LysoTracker red fluorescence intensity in serum starved WT vs. DKO MEFs for the indicated time points. (**D**) Representative trace for LysoTracker red fluorescence in WT and DKO MEFs serum-starved for 12 hr and treated with or without bafilomycin A1. Fluorescent intensity was continually measured for 40 min following the bafilomycin A1 addition. (**E**) Confocal images of WT and DKO MEFs treated the same as in panel 'D'. Scale bars = 10 μm. (**F**) Graph of cytosolic pH in WT and DKO MEFs in control serum-replete conditions (black) or serum-starved (red) conditions. (**G**) Confocal images of active cathepsin B (red) in WT and DKO MEFs in control or serum-starved state. Scale bars = 5 μm. All assays are an average, or are representative of three independent experiments. The error bars represent the standard error of the mean. *$p<0.05$ vs 0 serum-replete. Statistical significance was determined by student's t-test.

DOI: https://doi.org/10.7554/eLife.30543.008

suggest that exogenous expression of Bax does not target exclusively to mitochondrial membranes but that some targets lysosomes and associated downstream vesicles.

To more definitively determine if autophagic cell death is dependent on the localization of Bax to the lysosome we generated a lysosomal-targeted Bax construct consisting of full-length Bax fused to a targeting sequence from Lamp1. DKO MEFs were infected with a doxycycline (Dox)-inducible adenovirus containing this construct and then subjected to serum starvation or staurosporine, with and without Dox treatment. Cell death was significantly restored in the Dox treated, serum-starved DKO

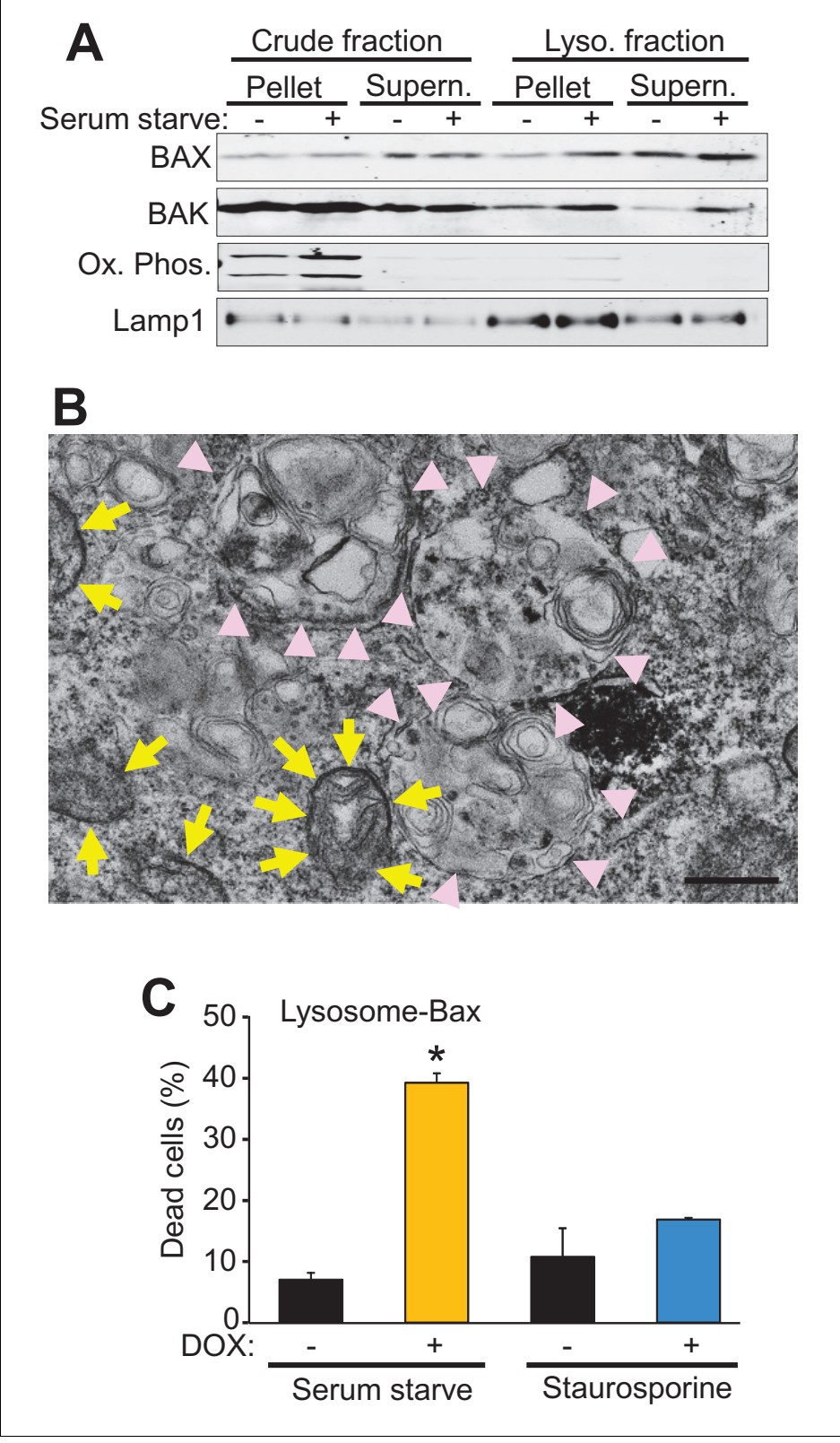

**Figure 4.** Bax and Bak localize to lysosomes and targeting Bax to lysosomes restores autophagic cell death in DKO MEFs. (**A**) Western blot analysis of the indicated protein fractions from starved and control WT MEFs for Bax, Bak, Lamp1 and the oxidative phosphorylation mitochondrial proteins UQCRC2 (upper band) and MTCO1 (lower band). (**B**) Transmission EM image of DKO MEFs treated with an adenovirus for mini SOG-Bax. The yellow

*Figure 4 continued on next page*

*Figure 4 continued*

arrows show electron dense mini-SOG-Bax within the mitochondrial membranes while the pink arrow heads show electron dense regions within the autolysosomes and lysosomes. Scale bar = 1 μm. (C) DKO MEFs infected with a Dox-inducible adenovirus for Bax containing the lysosomal targeting sequence of Lamp1 on its C-terminus. The infected cells were then treated with or without Dox to induce expression and then subjected to serum starvation to induce autophagic cell death, or treated with 200 nM staurosporine to induce apoptotic cell death. Cell death was determined by plasma membrane rupture. All assays are an average or are representative of three independent experiments. The error bars represent the standard error of the mean. *p<0.05 vs no Dox with students t-test.

DOI: https://doi.org/10.7554/eLife.30543.009

The following source data is available for figure 4:

**Source data 1.** Raw western gel images for *Figure 4A*.

DOI: https://doi.org/10.7554/eLife.30543.010

---

MEFs, but not in the Dox treated staurosporine treated DKO MEFs (*Figure 4C*). Staurosporine-induced apoptotic cell death requires mitochondrial Bax or Bak for cell death, and the fact that the lysosomal targeted Bax did not restore killing with this treatment indicates that the fusion protein successful targeted Bax to the lysosome but not the mitochondria. Together these data strongly support a mitochondrial-independent, lysosome-dependent role for Bax and Bak as a requisite mechanism underlying autophagic cell death.

## Lysosomal permeabilization restores autophagic cell death in Bax/Bak1 DKO MEFs

To further investigate the mechanistic basis for Bax/Bak-mediated autophagic cell death through lysosomal permeability we employed the lysosomotropic detergent O-methyl-serine dodecylamide hydrochloride (MSDH) that induces lysosomal membrane destabilization (*Li et al., 2000*). Serum-starved DKO MEFs loaded with LysoTracker red were shown to be resistant to loss of lysosomal permeability and loss of red fluorescence over 40 min of vehicle treatment, while addition of MSDH induced permeability and loss of fluorescence (*Figure 5A*). More importantly, restoration of LMP by MSDH treatment now rendered the DKO MEFs susceptible to serum starvation-induced autophagic cell death (*Figure 5B,C*). More specifically, pre-starvation for 24 hr in DKO MEFs followed by MSDH treatment for 12 hr restored killing yet MSDH alone had no effect in serum-replete medium (*Figure 5B*). In addition, simultaneous treatment with serum starvation and MSDH over 24 hr restored killing in DKO MEFs (*Figure 5C*). Upon ultrastructural analysis by transmission EM, lysosomal morphology in the starved DKO MEFs treated with MSDH was reminiscent of lysosomes from serum-starved WT MEFs (*Figure 5D*, arrows show large lysosomes/autolysosomes). Collectively these results suggest that autophagic cell death requires Bax or Bak to induce lysosomal membrane permeability in a process that is independent of the previously recognized apoptotic and regulated necrotic modes of action for these effectors at the level of the mitochondria.

## Discussion

The autophagic process normally serves a physiologic function to maintain cellular viability during times of starvation, although the prolongation of this starvation state eventually leads to cell death through a hypothesized active induction of apoptosis mediators (*Kroemer and Levine, 2008*). Here, we show that autophagic cell death in its terminal phases can be independent of classical apoptosis or mitochondrial-dependent regulated necrosis, with ultrastructural morphological characteristics specific to autophagy. Micro- and macroautophagy are well-known mediators of cellular health that protects from age-dependent disease (*Rubinsztein et al., 2011*), although there are selected acute disease states that require autophagic mediators to induce this unique form or cell death (*Muller et al., 2017*). For example, heart-specific Beclin 1 overexpressing transgenic mice with enhanced autophagy show cardiomyopathy while Beclin 1 heterozygous KOs mice with reduced autophagy are protected from pressure-overload stimulated cardiomyopathy collectively suggesting that augmented autophagy can lead to a degenerative like disorder in the heart (*Zhu et al., 2007*). Furthermore, genetic inhibition of the autophagic pathway is protective against neuronal and

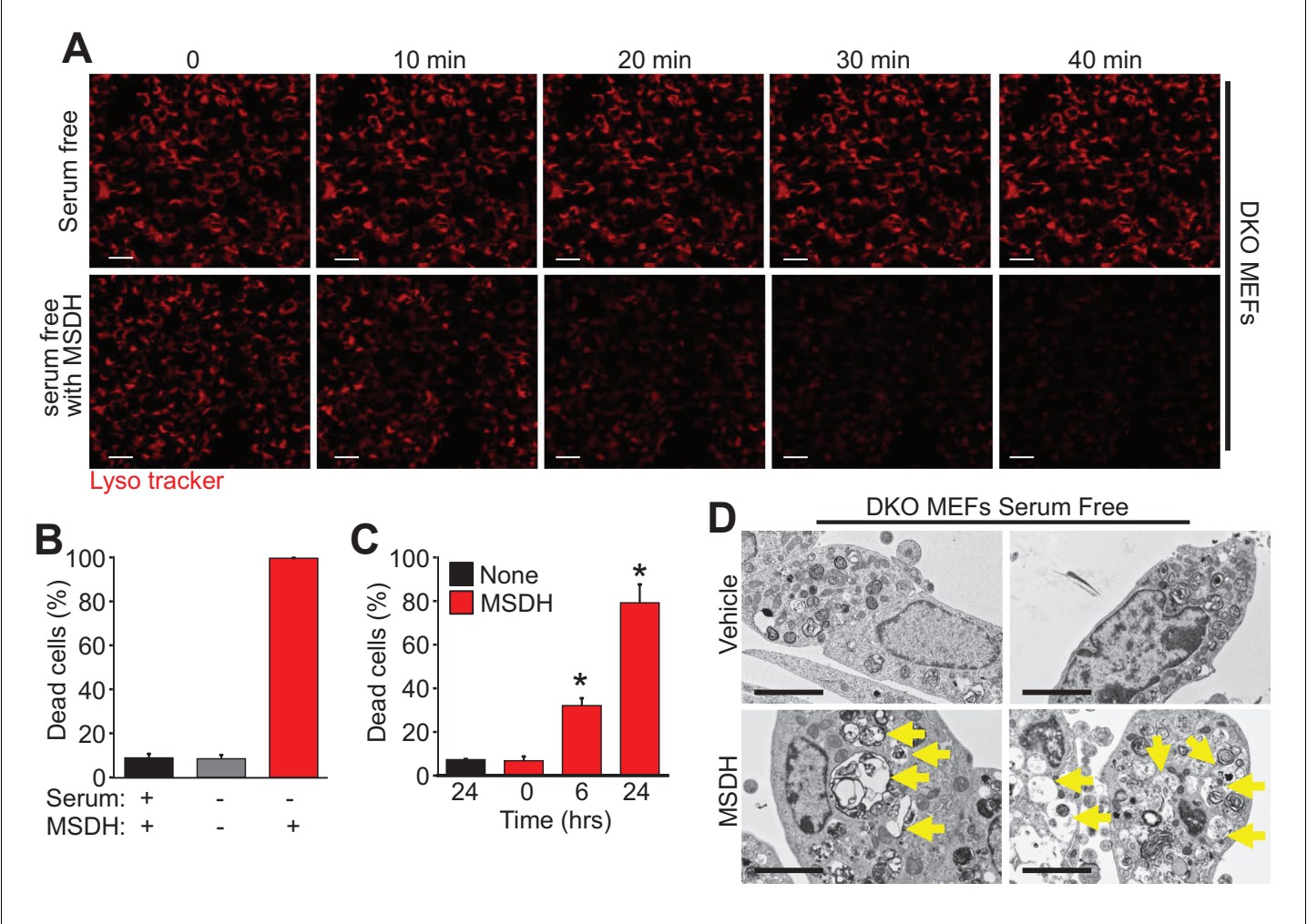

**Figure 5.** The lysosomotropic detergent MSDH restores autophagic death in *Bax/Bak1* DKO MEFs. (**A**) A time course of confocal images of DKO MEFs that were serum starved for 24 hr and then loaded with LysoTracker red, treated with or without 30 µM MSDH for the indicated times. Scale bars = 10 µm. (**B**) DKO MEFs with 24 hr serum starvation or control were treated with 30 µM MSDH and cell death was measured by loss of plasma membrane integrity 12 hr later. (**C**) Time course of serum-starved DKO MEFs treated with or without 30 µM MSDH for the indicated times. Cell death was measured as in panel 'B'. (**D**) Transmission EM images (magnification = 30,000 x) of 24 hr serum-starved DKO MEFs treated with or without 30 µM MSDH for 4 hr. Scale bars = 5 µm. All assays are an average or are representative of three independent experiments. The error bars represent the standard error of the mean. *p<0.05 vs untreated or time 0 with students t-test.

DOI: https://doi.org/10.7554/eLife.30543.011

cardiomyocyte cell death following ischemic injury (*Koike et al., 2008*; *Matsui et al., 2007*). These observations support the hypothesis that autophagic cell death can have a pathologic counterpart in vivo in selected tissues and disease states.

One unique and defining aspect of regulated autophagic cell death is the absolute requirement for LMP through Bax/Bak. Importantly, this unique mechanism of cell death is separate from micro- and macro-autophagy that are induced early in starvation to maintain cell viability, although the expansion of lysosomes and autolysosomes during these states likely primes the cells for transition to the more active autophagic killing process as these vesicular structures undergo permeabilization, which consequently lowers intracellular pH and/or releases cathepsins.

*Bax* and *Bak1* DKO MEFs are resistant to all three major forms of regulated cell death; apoptosis, necrosis and autophagy (*Wei et al., 2001*; *Karch et al., 2013*). Similar to their role in regulated necrosis, oligomerization of Bax and Bak are not required during the terminal process of autophagic cell death. Bax and Bak have been shown to lead to 2 states of permeability in membranes

(*Karch et al., 2013*; *Dejean et al., 2005*). In their non-oligomeric state they increase the permeability characteristics of the outer mitochondrial membrane and other lipid bilayers, while in their oligomeric state they greatly increase membrane permeability by forming large pores (*Karch et al., 2013*). Here, we show that during autophagic cell death Bax and Bak are localized to and mediate an increase in lysosomal permeability in their monomeric state, without being overtly activated, similar to how they increase the general permeability characteristics of the mitochondrial outer membrane (*Karch et al., 2013*). In isolated lipid bilayers Bax alone increases membrane permeability caharacteristics (*Karch et al., 2013*). Also, multiple Bcl-2 family members including Bax and Bak can act on channels to influence membrane permeability (*Shimizu et al., 1999*). In regards to LMP, it is unknown whether Bax/Bak act alone or with some other lysosomal membrane protein to increase permeability, although our results with MSDH suggest that it is simply the intrinsic permeability enhancing characteristics of these effectors that likely allows lysosomes to leak and/or rupture during the final phases of autophagic cell death. Indeed, gossypol restores autophagic killing in the absence of Bax/Bak by simply converting the anti-apoptotic Bcl-2 proteins into agents that increase membrane permeability like Bax/Bak.

Our results collectively place Bax and Bak at the nodal intersection between all three major forms of regulated cell death. Further elucidation of the crosstalk that takes place between these pathways through Bcl-2 family members has many diverse implications for medical applications. It will also be important to further characterize how Bax and Bak are selectively localized to the lysosome versus the mitochondria at baseline and with cell death inducing stimuli. It is remarkable that loss of Bax/Bak fully protect cells from serum starvation and terminal autophagy-dependent cell death, suggesting that inhibitors of monomeric Bax/Bak could be selectively used in preventing or reducing pathology and cell loss associated with selective degenerative disease states during aging while not inhibiting the protective effects ascribed to the autophagic process (*Muller et al., 2017*).

# Materials and methods

**Key resources table**

| Reagent type (species) or resource | Designation | Source or reference | Identifiers | Additional information |
|---|---|---|---|---|
| gene (Mus Musculus) | Bak1 | N/A | ID: 12018 | N/A |
| gene (Mus Musculus) | Bax | N/A | ID: 12028 | N/A |
| cell line (Mus Musculus) | Wild Type SV40 Mouse Embryonic Fibroblasts; WT MEFs | PMID: 11326099 | N/A | N/A |
| cell line (Mus Musculus) | Bax Bak1 double knockout SV40 Mouse Embryonic Fibroblasts; DKO MEFs | PMID: 11326099 | N/A | N/A |
| cell line (Mus Musculus) | DKO MEFs stably reconstitutied with wild type Bax; DKO-WT-Bax | PMID: 21255726 | N/A | N/A |
| cell line (Mus Musculus) | DKO MEFs stably reconstitutied with 92-94A mutant Bax; DKO-92-94A-Bax | PMID: 21255726 | N/A | N/A |
| cell line (Mus Musculus) | DKO MEFs stably reconstitutied with 63-65A mutant Bax; DKO-63-65A-Bax | PMID: 21255726 | N/A | N/A |
| antibody | Bax6A7 | Thermo Fisher Scinentific | MA5-14003 | 1 to 100 |
| antibody | TOM20 | Santa Cruz Biotechnology | sc-11415 | 1 to 100 |
| antibody | Lamp-1 | DSHB-University of Iowa | 1D4B | 1 to 200 and 1 to 1000 |
| antibody | Bax | Santa Cruz Biotechnologies | sc-493 | 1 to 300 |
| antibody | Bak | Millipore | 06–536 | 1 to 1000 |
| antibody | LC3I/II | Abcam | ab58610 | 1 to 500 |
| antibody | Bcl-2 | Santa Cruz Biotechnologies | sc-7382 | 1 to 300 |
| antibody | Lamp1 | Abcam | ab24170 | 1 to 500 |
| antibody | Cathepsin D | Novus | AF1029 | 1 to 800 |
| antibody | GAPDH | Fitzgerald | 10R-G109A | 1 to 25,000 |

*Continued on next page*

*Continued*

| Reagent type (species) or resource | Designation | Source or reference | Identifiers | Additional information |
|---|---|---|---|---|
| antibody | Total OXPHOS Rodent WB Antibody Cocktail | Abcam | ab110413 | 1 to 10,000 |
| antibody | αTubulin | Santa Cruz Biotechnologies | sc-8035 | 1 to 250 |
| recombinant DNA reagent | Lysosomal targeted Bax; Lyso-Bax | Bio Basic | N/A | special order |
| recombinant DNA reagent | MiniSOG-Bax | Bio Basic | N/A | special order |
| sequence-based reagent | N/A | N/A | N/A | N/A |
| peptide, recombinant protein | N/A | N/A | N/A | N/A |
| commercial assay or kit | Lysosome Isolation Kit | Sigma | LYSISO1 Sigma | N/A |
| commercial assay or kit | intracellular pH detection kit | Invitrogen | P35379 | N/A |
| commercial assay or kit | Muse Count and Viability assay | EMD Millipore | MCH100102 | N/A |
| chemical compound, drug | ABT-737 | Selleck Chemicals | S1002 | 20 µM |
| chemical compound, drug | Gossypol | Tocris | 1964 | 10 µM |
| chemical compound, drug | caspase inhibitor zVAD-fmk | Promega | G7232 | 20 µM |
| chemical compound, drug | (S)-O-methyl-serine dodecylamide hydrochloride; MSDH | Avanti Polar Lipids | 850546 | 30 µM |
| chemical compound, drug | staurosporine | Sigma-Aldrich | S6942 | 200 nM |
| chemical compound, drug | LysoTracker Red DND-99 | ThermoFisher Scientific | L7528 | 50 nM |
| chemical compound, drug | Rhodamine-123 | ThermoFisher Scientific | R-302 | 25 µM |
| chemical compound, drug | pHrodo Red AM Intracellular pH Indicator | ThermoFisher Scientific | P35372 | N/A |
| chemical compound, drug | Protease inhibitor cocktail | ThermoFisher Scientific | 88666 | N/A |
| software, algorithm | Excel 2013 | Microsoft Office | N/A | N/A |
| software, algorithm | Power Point 2013 | Microsoft Office | N/A | N/A |

## Tissue culture, cell lines, treatments, and analysis of cell death

All MEFs were cultured in IMDM medium (ThermoFisher Scientific, Waltham, MA. SH3022801) supplemented with 10% bovine growth serum (BGS, Fisher Scientific, SH3054103) and in the presence of penicillin-streptomycin (ThermoFisher Scientific, 30–002 CI), and nonessential amino acids (Invitrogen, Carlsbad CA. 11140–050). DKO MEFs stably expressing WT Bax or Bak or mutant Bax were previously described (*Hoppins et al., 2011*; *Kim et al., 2009*). At 80% confluency cells were subjected to serum starvation or amino acid depletion by replacing the 10% BGS-IMDM with 0% BGS-IMDM with the same supplements or replacing the media with DMEM lacking L-Arginine, L-Glutamine, and L-Lysine (Fisher Scientific, A1443101) supplemented with only penicillin-streptomycin. Before media was replaced, the cells were thoroughly washed three times with phosphate buffered saline (PBS). In addition to starvation a portion of cells were treated with 20 µM ABT-737 (Selleck Chemicals, Houston, TX. S1002), 10 µM Gossypol (Tocris, Bristol, United Kingdom. 1964), 20 µM caspase inhibitor zVAD-fmk (Promega, Madison, WI. G7232), or with 30 µM (S)-O-methyl-serine dodecylamide hydrochloride MSDH (Avanti Polar Lipids, Alabaster, AL.850546). As a positive control for apoptotic cell death MEFs were also treated with 200 nM staurosporine (Sigma-Aldrich, St Louis, MO. S6942) for 24 hr. A portion of MEFs were also infected with a previously described adenovirus for LC3-GFP (*Matsui et al., 2007*), a Lyso-Bax DOX inducible adenovirus, or mini SOG-Bax adenovirus. Cell death was determined by the Muse Count and Viability assay (EMD Millipore, Burlington, MA. MCH100102). Briefly, cells were trypsinized and washed twice and incubated with Muse Count and Viability reagent. The Muse Count and Viability Reagent is a two dye system where one dye labels all cells dead or alive while the other only labels dead cells due to loss of plasma membrane integrity. The cells were then quantified on a Muse cell analyzer (EMD Millipore) at 5000 counts per sample. All cell lines used were authenticated by western blot analysis for the proposed recombinant constructs by western blotting, or MEFs were confirmed to be absent of Bax/Bak1 by western

blotting, as well as blotting for the oligomerization dead mutants of Bax as authentication. No myco-plasma contamination was observed.

## Generation of adenoviral constructs

Lysosomal Bax (Lyso-Bax) was commercially generated (Bio Basic) by adding the lysosomal targeting sequence of Lamp1, GYQTI (ggc tat cag acc atc), to the C-terminal end of a full length Bax cDNA prior to the stop codon. The construct was then cloned into the Adeno-X Tet-On 3G Vector (Clontech, Moutnain View, CA. 631180) to produce purified adenovirus. The mini SOG-Bax construct was commercially generated (Bio Basic) by adding the mini SOG tag (*Shu et al., 2011b*) to the N-terminus of the full-length Bax cDNA minus the start codon. The construct was then cloned into the pShuttle-CMV vector to produce purified adenovirus.

## Electron and confocal microscopy

Electron microscopy was performed on fed and serum-starved WT and DKO MEFs. Prior to fixation, some cells were subjected to 1–4 days of serum starvation and/or treated with 30 μM MSDH for 4 hr. Samples were then fixed in glutaraldehyde and cacodylate, embedded in epoxy resin, sectioned, and counterstained with uranyl acetate and lead citrate. The sample preparation for mini-singlet oxygen generator (SOG) imaging by EM was a modified version of the protocol originally described by the Tsien lab (*Shu et al., 2011b*). Briefly, DKO MEFs were fixed with 1% glutaraldehyde in a 0.1 M phosphate buffer made with $Na_2HPO_4$ dibasic anhydrous (1.15 g/100 ml), pH 7.4, for 30–60 min on ice, then rinsed several times with 0.1 M phosphate buffer, pH 7.4. MEFs were then blocked with a buffer containing 50 mM glycine, 10 mM KCN, and 5 mM aminotriazole for 30 min on ice. Blocking buffer was then removed and replaced with 1 mg/mL 3,3′ diaminobenzidine (DAB) in 0.1 M phosphate buffer and incubated on ice for 15 min. Photooxidation was performed with a FITC filter set under illumination from a SOLA Light Engine (Lumencor, Beaverton, OR.) for 4 min with pure oxygen flow directly onto the cells until a light brown precipitate appeared. Cells were then washed with ice cold 0.1 M phosphate buffer for 2 min, five times. Cells were post-fixed with 1% OsO4 in 0.1 M phosphate buffer for 30 min on ice, washed two times withice cold 0.1 M phosphate buffer, and one time with distilled water. Cells were then stained with 2% aqueous uranyl acetate for 1 hr to overnight, before dehydration through an ethanol series (20%, 50%, 70%, 90%, 100%, 100%) for 2 min each, before a final rinse in anhydrous ethanol, followed by infiltration into a 1:1 ratio of epon and anhydrous ethanol for 30 min, then in 100% epon twice for 1 hr each, then into fresh resin and vacuum polymerized.

Live cell fluorescent microscopy was performed on WT and DKO MEFs. MEFs were plated on 14 mm glass bottom dishes (MatTek Corporation, Ashland, MA. p35G-0–10 C). At 80% confluency MEFs were subjected to serum starvation for the indicated time points. Prior to imaging, cells were loaded with LysoTracker red DND-99 (ThermoFisher Scientific, L7528) and/or Rhodamine-123 (ThermoFisher Scientific, R-302). 50 nM LysoTraker red and/or 25 μM Rhodamine-123, the latter of which was imaged with green channel settings, was added to the cells for 10 min followed by two washes with Hank's Balanced Salt Solution (HBSS) and then fresh media, minus the phenol red, either containing 10% or 0% BGS, was added. For the immunocytochemistry experiments, cells were plated on 2–well glass chamber slides and treated with the indicated treatment. Following treatment the cells were fixed in 4% PFA at room temperature for 20 min. Cells were then incubated in blocking buffer (5% goat serum, 2% BSA, and 0.1% Triton X-100 in PBS) for 20 min followed by overnight primary antibody incubation at 4°C. The cells were then washed 3 times with PBS and incubated at room temperature for 2 hr with the appropriate secondary antibodies. Cell images were obtained using an inverted Nikon A1R confocal microscope. The following antibodies were used: Bax6A7 (Thermo Fisher Scinentific, MA5-14003, 1:100), TOM20 (Santa Cruz Biotechnology, Santa Cruz, CA. sc-11415, 1:100), and Lamp-1 (DSHB-University of Iowa, Iowa City, IA. 1D4B, 1:100).

## Determination of intracellular pH

Intracellular pH was quantified in WT and DKO MEFs with and without serum starvation using the intracellular pH detection kit (Invitrogen, P35379) according to the manufacturer's instructions. Briefly, MEFs were plated on 96-well dishes under normal conditions and the next day they were given fresh media with or without 10% BGS. 24 hr later, cells were washed with HBSS, loaded with

pHrodo Red AM Intracellular pH Indicator (Thermo Fisher Scientific, P35372) in HBSS, then washed two times with HBSS and incubated at 37°C for 15 min and then read on a plate reader (BioTek, Winooski, VA.) with excitation at 560 nm and emission detection at 580 nm. The intracellular pH calibration buffer kit was used to by performing the assay as above with pH 4.5, 5.5, 6.5, or 7.5 calibration buffers to generate a standard curve of pHrodo Red fluorescence versus pH.

## Lysosomal fractionation and western blotting

Lysosomal fractionation was performed using the Lysosome Isolation Kit (Sigma, LYSISO1). Briefly, WT MEFs were plated on six large bioassay dishes (ThermoFisher Scientific, 166508). Half of the plates were subjected to serum starvation for 24 hr. Cells were then washed, trypsinized and pelleted. Following an additional wash, cells were suspended in extraction buffer and subjected to Dounce homogenization (Pestle B) followed by centrifugation (1000 x g for 10 min). The supernatant was collected and then centrifuged at 20,000 x g for 20 min, which generated a pelleted crude lysosomal fraction, from which an aliquot of both the pellet and supernatant was collected for western blot analysis. The remaining crude lysosomal fraction was placed on an Optiprep step gradient and ultra-centrifuged at 150,000 x g for 4 hr. The lysosomal fractions werecollected and treated with 8 mM $CaCl_2$ for 15 min to further purify the lysosomes from the remaining mitochondria or ER. Finally the samples were centrifuged at 5000 x g for 15 min and both the pellet and supernatant (purest lysosomal fraction) were collected for western blot analysis. Protease inhibitor cocktail (Thermo Fisher Scientific, 88666) was added to all samples. The MEFs and collected lysosomal pellets were homogenized in RadioImmunoPrecipitation Assay (RIPA) buffer containing protease inhibitor cocktails. After protein concentration was determined, SDS sample buffer was added to all samples and western blots were performed. For the Bak oligomerization assay WT MEFs were serum-starved for 48 hr or treated with 200 nM staurosporine and 20 µM zVAD-FMK for 24 hr, or left untreated and then homogenized in tris-buffered saline containing 1% n-Dodecyl-β-D-Maltoside and 1% digitonin. After the protein concentration was assessed NativePAGE sample buffer was added and a Western blot was performed. The following antibodies were used: Bax (Santa Cruz Biotechnologies, sc-493, 1:300), Bak (Millipore, Burlington, MA. 06–536, 1:1000), LC3I/II (Abcam, Cambridge, UK. ab58610, 1:500), Bcl-2 (Santa Cruz Biotechnologies, sc-7382, 1:300), Lamp1 (Abcam, ab24170, 1:500), Lamp-1 (DSHB-University of Iowa, 1D4B, 1:1000), Cathepsin D (Novus, Uppsala, Sweden. AF1029, 1:800), GAPDH (Fitzgerald, Acton, MA. 10R-G109A, 1:25,000), Total OXPHOS Rodent WB Antibody Cocktail (Abcam, ab110413, 1:10,000) αtubulin (Santa Cruz Biotechnologies, sc-8035, 1:250).

## Statistics

All results are presented as the mean ± SEM and all data were normally distributed. The statistical analysis performed was an unpaired two-tailed Student's t test using Excel 2013 and values were considered statistically significant when $p < 0.05$. No statistical analysis was used to predetermine sample size. The sample number of biological replicates for each experiment is indicated in the figure legends and no outlier data were excluded. The experiments were not generally blinded except for the data acquisition for the following figures: *Figure 1D*, *Figure 3A*, and *Figure 5D* (which were blinded).

## Acknowledgements

This work was supported by grants from the National Institutes of Health (JDM) and the Howard Hughes Medical Institute (JDM, and JK).

## Additional information

### Funding

| Funder | Grant reference number | Author |
| --- | --- | --- |
| Howard Hughes Medical Institute | Molkentin | Jeffery D Molkentin Jason Karch |
| National Institutes of Health | R01HL132831 | Jeffery D Molkentin |

The funders had no role in study design, data collection and interpretation, or the decision to submit the work for publication.

## Author contributions
Jason Karch, Conceptualization, Data curation, Investigation, Writing—original draft; Tobias G Schips, Bryan D Maliken, Matthew J Brody, Michelle A Sargent, Onur Kanisicak, Data curation; Jeffery D Molkentin, Conceptualization, Supervision, Funding acquisition, Writing—original draft, Project administration, Writing—review and editing

## Author ORCIDs
Jeffery D Molkentin (iD) http://orcid.org/0000-0002-3558-6529

## Decision letter and Author response
Decision letter https://doi.org/10.7554/eLife.30543.015
Author response https://doi.org/10.7554/eLife.30543.016

# Additional files

## Supplementary files
• Transparent reporting form
DOI: https://doi.org/10.7554/eLife.30543.012

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
