## [Decision Letter]

Thank you for submitting your article "Autophagic cell death is dependent on lysosomal membrane permeability through Bax and Bak" for consideration by *eLife*. Your article has been reviewed by two peer reviewers, and the evaluation has been overseen by Fiona Watt as the Senior and Reviewing Editor. The following individual involved in review of your submission has agreed to reveal his identity: Atan Gross (Reviewer #2).

The reviewers have discussed the reviews with one another and the Reviewing Editor has drafted this decision to help you prepare a revised submission.

Summary:

In this paper, Karch et al. demonstrate that Bax/Bak double KO (DKO) MEFs are resistant to autophagy (serum starvation)-induced cell death. The authors demonstrate that Bax, in its monomeric form and specifically targeted to the lysosome, is sufficient to increase lysosomal membrane permeability and to restore autophagic cell death in the DKO MEFs. In addition, increasing lysosomal permeability using a lysomotropic detergent also rescued autophagic cell death in the DKO MEFs. Taken together, this is an important study that shows that as like in apoptosis and necrosis Bax/Bak play a critical role in autophagic cell death by regulating lysosome membrane permeability.

Essential revisions:

1) "Autophagic cell death" is not a well-defined term. A detailed description of the morphology of dead cells will help the readers understand better. The cell death assay mentioned in the Materials and methods section is the Muse Count & Viability assay, which uses a cell membrane impermeable dye to mark dead cells. Can the authors provide direct evidence that the dead cells have a compromised plasma membrane?

2) The claim of "autophagic cell death" is not sufficiently demonstrated. There is autophagy in the serum-starved cells and there is cell death. But the authors did not provide sufficient evidence that the cell death induced by serum starvation is driven by autophagy, not just associated with autophagy. It is well known that serum starvation will induce autophagy, apoptosis, and non-apoptotic cell death. To demonstrate the cell death observed in WT MEF cells, the following experiments will be helpful. (A) It will be more convincing if the authors could use a more defined autophagy inducer, such as amino acid deprivation to induce cell death. (B) In the serum-starvation induced cell death system, it will be more convincing if the authors could show inhibition with autophagy inhibitor such as 3-MA, or by loss-of-function of autophagic genes (ATG5/6 etc.) could block cell death.

3) Figure 2: The authors show that oligomerization-deficient mutants of Bax restore autophagic cell death in the DKO MEFs. Based on these findings, they conclude that "These results indicate that Bax/Bak activation and oligomerization are not required for serum starvation-induced cell death". The fact that Bax oligomerization does not seem to be important for this form of cell death does not allow the conclusion that Bax does not need to be activated. Does serum-starvation induce the insertion of Bax into the lysosome membrane to become an integral membrane protein? Does Bax go through a conformational change (e.g., N-term exposure)? Does it homo-oligomerize in the lysosome membrane? In addition, in order to compare the effect of WT Bax and Bax mutants, expression level of these proteins need to be shown by Western blotting.

4) Figure 4. The authors imply that lysosomes are permeabilized in WT MEF cells after serum starvation. The authors need to show more clearly, perhaps using immunofluorescence, that serum-starvation indeed triggers the translocation of Bax to the lysosome membrane. How much cytosolic Bax is translocated to the lysosome as compared to the mitochondria following serum-starvation? Do other forms of cell death (apoptosis, necrosis) also induce Bax translocation to the lysosome?

Other points:

1) In Figure 1, The GFP-LC3 dots are not very obvious. It will help to provide enlarged higher-resolution pictures and quantify the dots.

2) In Figure 3, after 3 days of starvation, active cathepsin B staining was striking in the WT MEF cells. But the image could not demonstrate the active cathepsin B is in the lysosomes or cytosol. This could be resolved by GFP-LAMP1 viral transduction. An important point of the manuscript is that LMP by Bax/Bak is required for autophagic cell death. But the evidence for lysosomes content spilling into the cytosol is missing. The lysotracker staining going down could be because the ability to maintain lysosome pH is compromised in serum-starved cells, not LMP.

3) Figure 4: The message in this picture is not clear since miniSOG-Bax seems to be expressed in many regions of the cell (not only in the mitochondria and lysosome as depicted by the arrows).

4) Figure 4: The authors should show that lysosome-Bax is exclusively targeted to the lysosome. Can a mito-targeted Bax also rescue serum starvation-induced cell death in DKO MEFs?

---

## [Author Response]

Essential revisions:1) "Autophagic cell death" is not a well-defined term. A detailed description of the morphology of dead cells will help the readers understand better. The cell death assay mentioned in the Materials and methods section is the Muse Count & Viability assay, which uses a cell membrane impermeable dye to mark dead cells. Can the authors provide direct evidence that the dead cells have a compromised plasma membrane?

We agree the term “Autophagic cell death” has been used in a confusing way in the literature and there is little consensus, as the original intended use was signifying a protective mechanism for a cell to prolong life. However, the field has generally usurped this term to refer to the eventual death of starved cells that were in a prolonged period of protection due to "internal combustion". To provide better clarity on our use of the term “autophagic cell death” we have added a description of how this process is morphologically unique from apoptosis and necrosis to the Introduction of the manuscript. As further definitional support, we also assess plasma membrane rupture by using a membrane impermeable dye and we have used EM pictures to show the ultrastructural hallmarks of this protracted type of cell death. In the revised manuscript we also add EM pictures of 4 day serum starved WT and DKO MEFs to Figure 1. Indeed, these pictures display direct plasma membrane rupture in the WT and not in the DKO MEFs.

2) The claim of "autophagic cell death" is not sufficiently demonstrated. There is autophagy in the serum-starved cells and there is cell death. But the authors did not provide sufficient evidence that the cell death induced by serum starvation is driven by autophagy, not just associated with autophagy. It is well known that serum starvation will induce autophagy, apoptosis, and non-apoptotic cell death. To demonstrate the cell death observed in WT MEF cells, the following experiments will be helpful. (A) It will be more convincing if the authors could use a more defined autophagy inducer, such as amino acid deprivation to induce cell death. (B) In the serum-starvation induced cell death system, it will be more convincing if the authors could show inhibition with autophagy inhibitor such as 3-MA, or by loss-of-function of autophagic genes (ATG5/6 etc.) could block cell death.

We agree that it would be beneficial to test whether Bax and Bak play similar roles in other models of autophagic induced cell death like amino acid depravation. Indeed, in the revised manuscript we show that loss of Bax and Bak rendered MEFs protective from amino acid depravation-induced autophagic cell death (Figure 1—figure supplement 1). In addition, we also examined the lysosomes of amino acid deprived WT and DKO MEFs using lyso-tracker Red and we observed that the intensity of the stain was enhanced in the deprived DKO MEF vs the WT MEFs (Figure 1—figure supplement 1), very similar to what we report with serum starvation. As stated above, we believe that the autophagic process is at first a protective process rather than maladaptive during “autophagic cell death”, which explains our results in attempting to perform this experiment with 3-MA. Indeed, use of 3-MA is serum-starved WT MEFs enhanced cell death, but extent of death and earlier onset (data not shown). These results are consistent with previous findings in the literature (Shimizu et al., 2004) hence we did not include such a protocol as it would be confusing to the readership.

3) Figure 2: The authors show that oligomerization-deficient mutants of Bax restore autophagic cell death in the DKO MEFs. Based on these findings, they conclude that "These results indicate that Bax/Bak activation and oligomerization are not required for serum starvation-induced cell death". The fact that Bax oligomerization does not seem to be important for this form of cell death does not allow the conclusion that Bax does not need to be activated. Does serum-starvation induce the insertion of Bax into the lysosome membrane to become an integral membrane protein? Does Bax go through a conformational change (e.g., N-term exposure)? Does it homo-oligomerize in the lysosome membrane? In addition, in order to compare the effect of WT Bax and Bax mutants, expression level of these proteins need to be shown by Western blotting.

The reviewers are correct that just because DKO MEFs reconstituted with oligomeric dead mutants of Bax can restore serum-deprived cell death does not mean that endogenous Bax activation, membrane insertion and oligomerization is not happening in starved WT MEFs. In the revised manuscript we address these issues. Using an antibody that specifically recognizes active Bax (N-term exposure) we determined that serum starved MEFs do not have a detectable amount of active Bax, while staurosporine strongly gives reactivity with this antibody (Figure 2—figure supplement 1). In addition, since Bax does not become activated with this stimuli it would be unlikely that a detectable translocation event would occur. Indeed, using DKO MEFs expressing GFP fused to Bax we determined that under serum deprivation no noticeable translocation of Bax occurs and it remains mainly cytosolic, yet staurosporine strongly induces translocation and aggregation (Figure 2—figure supplement 1). Furthermore, we measured the oligomeric state of Bak under serum deprivation and did not detect an increase in Bak oligomers (Figure 2—figure supplement 1). In conclusion, unlike apoptosis, “autophagic cell death" does not require the activation and oligomerization of Bax and Bak but it does require their expression, which is similar to our previous mechanistic findings on regulated necrotic cell death at the level of the mitochondria through monomeric Bax or Bak, exclusively. In addition, we have added the requested Western blots to Figure 2—figure supplement 1 and show that the expression levels of WT Bax is nearly identical to the mutant versions (important control).

4) Figure 4. The authors imply that lysosomes are permeabilized in WT MEF cells after serum starvation. The authors need to show more clearly, perhaps using immunofluorescence, that serum-starvation indeed triggers the translocation of Bax to the lysosome membrane. How much cytosolic Bax is translocated to the lysosome as compared to the mitochondria following serum-starvation? Do other forms of cell death (apoptosis, necrosis) also induce Bax translocation to the lysosome?

As stated above we did not detect appreciable Bax translocation to the lysosome during serum starvation by immunofluorescence, as we believe that the monomeric form of Bax already in the lysosome it fully sufficient to mediate permeability. Indeed, Figure 4 shows by cellular fractionation and Western blotting that both Bax and Bak are part of the lysosome at baseline, with even a bit more following serum starvation. Currently, we are unsure as to how Bax and Bak get to the lysosome. One idea is that it is through mitochondrial membrane budding as has been previously suggested to aid in lysosomal formation (Soubannier et al., 2012). Another idea supported by the literature is that Bax and possibly Bak can be found in most intracellular membrane systems (endoplasmic reticulum) in the cell at a small but significant level. However, as far as other forms of cell death are concerned, we used staurosporine as an apoptotic control for the previously mentioned Bax translocation immunofluorescent experiments and observed Bax translocation to the mitochondria and not to the lysosome as expected (Figure 2—figure supplement 1). Hence, the mitochondria are still the primary mechanistic site for active and oligomerized Bax and Bak with dedicated docking proteins in this organelles outer membrane (VDAC).